# Comparison of Extracorporeal Shockwave Therapy with Non-Steroid Anti-Inflammatory Drugs and Intra-Articular Hyaluronic Acid Injection for Early Osteoarthritis of the Knees

**DOI:** 10.3390/biomedicines10020202

**Published:** 2022-01-18

**Authors:** Shun-Wun Jhan, Ching-Jen Wang, Kuan-Ting Wu, Ka-Kit Siu, Jih-Yang Ko, Wen-Chiung Huang, Wen-Yi Chou, Jai-Hong Cheng

**Affiliations:** 1Department of Orthopedic Surgery, Kaohsiung Chang Gung Memorial Hospital and Chang Gung University College of Medicine, Kaohsiung 833, Taiwan; b9502077@cgmh.org.tw (S.-W.J.); cjwang1211@gmail.com (C.-J.W.); enemy7523@gmail.com (K.-T.W.); kojy@cgmh.org.tw (J.-Y.K.); 2Center for Shockwave Medicine and Tissue Engineering, Kaohsiung Chang Gung Memorial Hospital and Chang Gung University College of Medicine, Kaohsiung 833, Taiwan; ay71004@gmail.com; 3Park One International Hospital, Kaohsiung 833, Taiwan; michaelskk@gmail.com; 4Medical Research, Kaohsiung Chang Gung Memorial Hospital and Chang Gung University College of Medicine, Kaohsiung 833, Taiwan

**Keywords:** knee, osteoarthritis, extracorporeal shockwave therapy, hyaluronic acid, non-steroid anti-inflammatory drugs

## Abstract

Conservative treatments for early osteoarthritis (OA) of the knee included the use of non-steroid anti-inflammatory drugs (NSAIDs) and intra-articular hyaluronic acid (HA) injection. Recently, several animal studies reported that extracorporeal shockwave therapy (ESWT) demonstrated chondroprotective effects on knee OA. The present study compared the efficacy of oral NSAIDs, HA injection, and noninvasive ESWT for early OA of the knee. Forty-five patients with early knee OA were randomized into three groups. NSAIDs group received celecoxib 200 mg daily for 3 weeks. HA group received intra-articular injection of HA once a week for 3 weeks. ESWT group received ESWT for 3 sessions at bi-weekly interval. All patients were followed up for one year. Evaluations included the visual analogue scale (VAS) score, serum enzyme-linked immunosorbent assay (ELISA), plain radiography, dual-energy X-ray absorptiometry (DEXA), and magnetic resonance imaging (MRI). In addition, the functional scores were performed including, WOMAC (Western Ontario and McMaster Universities Arthritis Index) score, KOOS (knee injury and osteoarthritis outcome) score, and IKDC (International Knee Documentation Committee) score. All three groups showed significant improvement in VAS and functional scores as well as in the collected one-year follow-up data after treatments. ESWT group had better pain relief than NSAIDs and HA groups. ESWT group had better therapeutic effects in the functional scores than NSAIDs and HA groups. The bone mineral density (BMD) of proximal tibia is significantly increased after ESWT than others. In the serum ELISA, ESWT inhibited the expression of COMP in knee OA patients as compared with NSAIDs and HA groups. The parameters of MRI showed no significant differences between three groups after treatments. ESWT and intra-articular HA injection showed comparable results than NSAIDs. ESWT was superior in pain relief than HA and NSAIDs. The results demonstrated that ESWT was an effective and alternative therapy than HA and NSAIDs for early osteoarthritis of the knees.

## 1. Introduction

Osteoarthritis (OA) of the knee is a common orthopedic disorder that causes pain, stiffness, and functional disability in daily activities [1]. Osteoarthritis affects medial compartment of the knee mostly. The inflammation of knee OA was induced in the synovial membrane and articular cartilage after injury. Many inflammatory cytokines (interleukin-1β, IL-1β; interleukin-6, IL-6; and tumor necrosis factor-α, TNF-α) and pathological factors (matrix metalloproteinase-13, MMP-13; vascular endothelial growth factor, VEGF) were released to the synovial fluid of knee OA [2]. In addition, inflammation-induced transforming growth factor beta (TGFβ), connective tissue growth factor (CTGF), bone morphogenetic protein-2 (BMP-2), BMP-5, and BMP-6 caused synovial and articular cartilage fibrosis and death in severe knee OA [3,4]. The goals of the treatments which include conservative and surgical treatments are pain relief, improvement of functional status and quality of life in patients with knee OA. Conservative treatments include using non-steroid anti-inflammatory drugs (NSAIDs), body weight reduction, knee brace, physical therapy, modification of activity level, intra-articular hyaluronic acid injection, and intra-articular platelet-rich plasma injection [5,6]. For example, platelet-rich plasma (PRP) is an autologous concentration of a patient’s own platelets, and can be easily prepared and injected into knee OA of patients by intra-articular injection. A clearer understanding of the mechanism of PRP can help in understanding anti-inflammation, stimulated chondrocyte proliferation, and tissue regeneration for OA in clinical trial and animal studies [7]. Surgical treatments include arthroscopic debridement and release, osteotomy and/or joint replacement. In addition, a new method named osteo-core plasty is a minimally invasive procedure to use for subchondral bone marrow lesion of knee OA [8]. Although joint arthroplasty is a mature and delicate surgery, knee joint preserving treatment should be considered in early stage osteoarthritis.

Hyaluronic acid (HA) is a natural polysaccharide, which acts as an important structural element of skin, subcutaneous tissue, and connective tissues [9]. HA can counter balance the intra-articular mechanical stress, and then protect the synovial tissue and articular cartilage [9]. In addition, HA is indicated for patients who have failed to respond to physical therapy or oral analgesics. A meta-analysis reported positive effects with intra-articular HA injection with limited time effect for up to six months [10].

OA had been considered an articular cartilage disease. However, many studies had shown that subchondral bone plays an important role that causes secondary articular cartilage change [11,12]. The increase in subchondral bone stiffness decreases the ability to scatter the loading forces within the knee joint, which then increases the force loaded on articular cartilage. Therefore, the cartilage damage and progress of OA accelerated over time [13]. The focus of treatment in early knee OA have shifted from the articular cartilage to the subchondral bone. Recent studies demonstrated that application of ESWT to the subchondral bone in the medial tibia condyle showed regression of early knee OA in rats [14].

Extracorporeal shockwave therapy (ESWT) has attracted intention and interest in musculoskeletal disorder for more than three decades. Shockwaves are high amplitude sound waves, generated by a high-voltage condenser spark discharge and then focused at the diseases area through an elliptical reflector [15,16]. Shockwaves can trigger biologic response to target tissue by inducing anti-inflammation, cell proliferation, and neovascularization, and then resulting in tissue regeneration and repair [15].

ESWT has shown effectiveness in the regression of early OA of the knee associated with decreased cartilage degradation and improves the subchondral bone remodeling in rats [17]. Several studies reported chondroprotective effect of ESWT in prevention and regression of OA of different joints in animals [18,19,20,21,22]. Dr. Dahlberg showed that ESWT significantly increased peak vertical force, lameness, and range of motion as compared to the control group in dogs [18]. Dr. Frisbie reported that ESWT significantly improved the degrees of lameness in horse [19]. Dr. Ochiai showed that ESWT is a useful treatment for knee OA with improvement in walking ability and reduction of calcitonin gene-related peptide in dorsal root ganglion neurons innervating the knee [21]. Dr. Wang reported that application of ESWT to the subchondral bone of the knee significantly increased bone volume and trabecular number and reduced bone porosity [23]. In addition, ESWT significantly increases the proliferating cell nuclear antigen positive cartilage cells, reaching a chondroprotective effect.

However, most studies used animal models to investigate the effect of ESWT [14,18,19,20,21,22]. There is still paucity in study to evaluate the effectiveness of ESWT in the treatment of early knee OA of the knee in human subjects. The present study aimed to investigate the efficacy of ESWT in early knee OA of human subjects by comparison with intra-articular hyaluronic acid injection and oral NSAIDs.

## 2. Materials and Methods

### 2.1. The Study Design

The CONSORT flow diagram is shown in Figure 1. Forty-five patients with fifty knees were randomized into three groups. The demographic data are shown in Table 1.

NSAIDs group consisted of 15 patients (17 knees) who were treated with oral administration of NSAIDs (celecoxib) (Pfizer Inc., Brooklyn, NY, USA) with the standard dose (200 mg daily) for three weeks. The mean age was 49 years old (range: 30–63 years). Patients with a history of gastrointestinal ulcers, renal and liver function failure, coagulopathy, and allergy were excluded.

HA group consists of 15 patients with 17 knees treated with three intra-articular hyaluronic acid injections weekly. The mean age was 52 years old (range: 33–64 years). The “hya-joint” hyaluronic acid was manufactured by SciVision Biotech Inc., Kaohsiung, Taiwan in a pre-packed 2.5 mL syringe ready for use.

ESWT group consists of 15 patients (16 knees) who were treated with the application of 3000 impulses of ESWT at energy level 4 (0.22 mJ/mm^2^ energy flux density) per session for three sessions at a bi-weekly interval. The mean age was 54 years old (range: 40–60 years).

All patients were allowed to resume normal activities of daily living, physiotherapy and the use of knee brace, and modification of activity levels.

### 2.2. The Patients

The study is a single-center, single-blind randomized clinical trial and approved by the Institutional Review Board of Chang Gung Memorial Hospital (IRB number: 101-3485A). Early knee OA patients, defined as Kellgren-Lawrence classification grade I and grade II, were included (Figure 2a) [24]. The radiographic feature of Kellgren-Lawrence grade I was minimal joint space narrowing and minute osteophyte. The radiographic feature of Kellgren-Lawrence grade II was possible mild joint space narrowing and definite osteophyte formation. Patients with symptomatic medial compartment osteoarthritis were enrolled, and those with lateral compartment OA were excluded. The sample size of *n* = 45 was estimated by Power analysis software (G*Power, version 3.1.9.2) with the post hoc test (two-tailed test [α, 0.05; β, 0.467]) (15 participants per group). The output data are including noncentrality parameter (5.866), critical t (2.022), Df (39.107), and Power (1-β err prob) (>0.9).

### 2.3. Enzyme-Linked Immunosorbent Assay

About 10 mL of blood was collected from patients and the serum was stored at −80 °C for use. Serum sample evaluation of our patients using serum ELISA and clinical examinations were carried out before treatment (week 0) and after at 48 weeks one-year follow-up. The serum levels of cartilage oligometric protein (COMP) (DCMP0, R&D Systems, Minneapolis, MN, USA), C-telopeptide of type II collagen (CTX-II) (CSB-E14328h, Cusabio Biotech Co., College Park, MD, USA), alkaline phosphatase (ALK-p) (DY370-05, R&D Systems, Minneapolis, MN, USA), osteocalcin (BMS2020INST, Gibco Invitrogen, Carlsbad, CA, USA), and insulin-like growth factor 1 (IGF-1) (DG100B, R&D Systems, Minneapolis, MN, USA) were measured by ELISA kit.

### 2.4. The Functional Scores

The clinical examinations included VAS (visual analogue scale) pain score, WOMAC (Western Ontario and McMaster Universities Arthritis Index) score, KOOS (knee injury and osteoarthritis outcome) score, and IKDC (international knee documentation committee) score [25].

### 2.5. The Plain Radiographies and MRI

The plain radiographies of the affected knee were obtained at 0, 12, 24, and 48 weeks. Dual-energy X-ray absorptiometry and MRI were performed at 0 and 48 weeks. The plain radiographies were used to examine the knee alignment, spur formation, joint space narrowing, Insall-salvati index, sulcus angle, and congruence angle. Dual-energy X-ray absorptiometry was used to measure the bone density of proximal tibia and distal femur (Figure 3). MRI was used to evaluate the cartilage thickening and degradation, and the patterns of subchondral trabecular bone [26,27]. Whole-organ MRI scoring method (WORMS) was applied to assess the knee joint with osteoarthritis change which included parameters such as articular cartilage integrity, subarticular bone marrow abnormality, subarticular cysts, subarticular bone attrition, marginal osteophytes, medial and lateral meniscal integrity, anterior and posterior cruciate ligament integrity, medial and lateral collateral ligament integrity, synovitis/effusion, intraarticular loose bodies, and periarticular cysts/bursitis [28].

### 2.6. Shockwave Application

The source of shockwaves was an Orthospec device (Medispec, Yehud, Isreal). The shockwaves were applied on the subchondral bone of the medial tibia condyle of the knee at 2.0 cm below the joint line in anteroposterior view and 2.0 cm from the medial skin edge in lateral view (Figure 2b). About 3000 impulses of shockwaves at energy level 4 (equivalent to 0.22 mJ/mm^2^ energy flux density) were delivered to the subchondral bone of the medial tibia condyle. Ice packing was applied immediately after the ESWT. Local reactions such as redness, swelling, and neurovascular complication were checked closely.

### 2.7. Statistical Analysis

SPSS ver. 17.0 (SPSS Inc., Chicago, IL, USA) was used in statistical analysis. The data were expressed as mean ± SD. A normality test of each variable was performed using Kolmogorov–Smirnova test. In variables subject to a normal distribution, Student’s *t*-test was used for comparison. In nonparametric variables, the Wilcoxon Signed Ranks test was used for intra-group evaluation and Mann–Whitney U test was used to compare the differences between the three groups (NSAIDs and HA, HA and ESWT, NSAIDs and ESWT). *p* values less than 0.05 were accepted as statistically significant.

## 3. Results

### 3.1. Comparison of the VAS and Functional Scores before and after NSAIDs, HA, and ESWT Treatments

Clinical outcomes were revealed and analysis before and after treatments is shown in Table 2. The three groups, NSAIDs, HA, and ESWT showed significant VAS score improvement after treatments. There was no significant difference in pain relief between NSAIDs and HA groups. The ESWT group had significant superiority in pain relief than HA (*p* = 0.015) and NSAIDs (*p* = 0.015) groups.

In the functional scores, the NASIDs group had significant improvements in KOOS (*p* = 0.037) and WOMAC (*p* = 0.005) scores but no significance in IKDC score after treatment. In addition, HA group was with significant improvements in KOOS (*p* = 0.006), WOMAC (*p* = 0.006), and IKDC (*p* = 0.005) scores after treatment. Further, ESWT group had significant improvements in KOOS (*p* = 0.001), WOMAC (*p* = 0.001), and IKDC (*p* = 0.001) scores at the end of the follow-up. The functional score improvement in KOOS and WOMAC scores after treatment was better in ESWT group, comparing with NSAIDs and HA groups (*p* < 0.05), and the difference was more prominent when comparing with NSAIDs group (*p* < 0.01). However, there was no significant difference in IKDC scores by comparing the three treatment groups with each other.

### 3.2. The Time Chasing and Tendency of VAS and Functional Scores after NSAIDs, HA, and ESWT Treatments

The trends of improvements in VAS, KOOS, WOMAC, and IKDC scores at pre-treatment, 2nd, 4th, 12th, 24th, and 48th week post-treatment are shown in Figure 3. The data illustrated that the VAS scores improved significantly at 4th week after treatments and was continuous to the end of the follow-up at 48th week in all three groups. The VAS scores of ESWT group were improved continuously for three months after treatment better than NSAIDs and HA groups.

The HA and ESWT groups had significant improvements from 4th to 48th weeks after treatments in KOOS, WOMAC, and IKDC scores compared with pretreatment and NSAIDs group. The NSAIDs group had substantial improvements from 4th to 48th weeks after treatments in KOOS and WOMAC scores but no significant improvement in IKDC score.

### 3.3. The Results of Enzyme-Linked Immunosorbent Assay

The data of ELISA analysis are shown in Table 3. The serum levels of ALK-P, osteoclacin, COMP, IGF-1, and CTX-II were measured by ELISA assay. The results showed no significant differences before and after treatment in all three groups. However, we observed that the level of serum COMP was still increased in NSAIDs (no significance) and HA (*p* = 0.024) groups after treatment. The level of serum COMP was almost the same in ESWT group before and after treatment. ESWT group prevented the increasing of COMP in the patient with knee OA after one-year follow-up treatment. Finally, there was almost no significant difference between three groups after treatment in the ELISA analysis except the COMP in HA group (*p* < 0.05).

### 3.4. The Analysis of Dual-Energy X-ray Absorptiometry and MRI

The bone mineral density (BMD) was measured and is shown in Figure 4. The BMD significantly increased in proximal tibia only after ESWT treatment (*p* = 0.004). Compared with HA group, the ESWT showed significant increase in BMD of the proximal tibia (*p* = 0.037).

There were no discernible differences of MRI parameters noticed in all three groups before and after treatment (data not shown).

## 4. Discussion

The principal findings of this study confirmed that the ESWT was an effective alternative for patients with symptomatic early OA. ESWT showed a significant progression in pain reduction and functional improvement over NSAIDs. Besides, ESWT had the same therapeutic effect as HA in functional improvement.

There are many studies that published the positive effect of shockwave on knee OA in animal models. Wang et al. have mentioned that shockwave has a chondroprotective effect associated with improvement in subchondral bone remodeling [17,29]. Because of the positive outcomes of ESWT in animal models, more and more research investigated the efficacy of ESWT on OA knees in human subjects. ESWT is a noninvasive procedure without significant complications. Several studies have mentioned the positive effect of ESWT on the treatment of knee osteoarthritis. Dr. Zhao reported that ESWT reduces pain and improves function in patients with knee osteoarthritis [30]. Dr. Li disclosed that ESWT is more effective in the Numeric Rating Scale and WOMAC score than laser therapy without adverse events [31]. Dr. Zhong compared the treatment of ESWT with placebo on knee OA, and the VAS, WOMAC, Lequesne index are superior in the ESWT group [32]. Dr. Uysal also reported the radial ESWT has statistically significant superior improvement than sham ESWT in all parameters, including VAS score, knee ROM, 20-m walk test, and Lequesne’s disability scores [33]. In our study, the ESWT group improved VAS, KOOS, WOMAC, and IKDC score statistically significantly one month after treatment, and the effect lasted for one year. There are some hypotheses about the mechanism of pain reduction after ESWT treatments. Bone marrow edema is a painful finding in patients with knee OA, and it may be the major source of knee pain. Dr. Kang demonstrated that ESWT is an effective and noninvasive treatment to reduce the painful bone marrow edema and shorten the disease course [34]. However, in our study, there was no significant change in MRI of the knee after shockwave treatment. All our patients had grade I or grade II knee osteoarthritis, so the bone marrow abnormality accounted for only 32.7% of all patients, and most were mild abnormalities. Because of the relative lower grade OA and low percentage bone marrow edema, there was no significant change after ESWT.

Dr. Xu compared the effect of ESWT with the effect of NSAIDs on OA knees. ESWT has the potential to reduce pain and to improve knee function. However, there is no statistical difference in VAS and WOMAC between ESWT and NSAIDs groups [35]. However, in our study, the ESWT group had more improvement in VAS score than the NSAIDs group. The NSAIDs group had no significant improvement in IKDC score after treatment and the ESWT group had significant improvement in all functional scores one month after treatment in our study. We applied focused ESWT in our patients and the therapeutic point was subchondral bone. The subchondral bone plays an important role in treatment of early knee OA [14]. Dr. Xu used radial ESWT and their therapeutic point was tender point of knee joints [34]. These differences might explain the different results between experiments. We can conclude that ESWT was superior to NSAIDs in treating patients with OA knees. In some patients with underlying diseases such as gastric ulcers, chronic kidney diseases, and cardiac vascular diseases, the ESWT is a better option than NSAIDs owing to the contraindications.

Dr. Lee compared the effect of ESWT versus HA for treatment of early knee OA. The results show that the score of VAS, WOMAC, Lequesne index, 40-m fast-paced walk test has significant improvement after treatment in both groups. There is no statistically significant difference between groups [36]. In the study, ESWT group has more improvements in VAS, KOOS, and WOMAC than HA group. Considering patients who have trypanophobia or refusion of invasive procedures, ESWT is an alternative treatment. The intra-articular HA injection has some complications reported [37,38]. The most common adverse effects were the injection-site reactions, such as, edema, pain, erythema, itching, and ecchymosis. Other complications of HA injection include hypersensitivity reactions, infections, and vascular occlusion. By contrast, there are rare complications associated with ESWT in OA knees [39]. Regarding the treatment in OA knees, ESWT is less invasive and has fewer complications than HA injection. When we evaluated the trend of VAS score improvement, only ESWT showed the continuous positive effect three months after treatment. Regarding pain relief outcomes, the ESWT could be a more durable treatment option than NSAIDs and HA.

Many studies have revealed that the ESWT posed the physical and chemical mechanical transduction with biological responses to achieve therapeutic purpose in biological tissue [16,17,40]. ESWT up-regulated the vWF, VEGF, BMP-2, osteocalcin, eNOS, TGF-β1, VEGF, and PCNA to induce anti-inflammation and neovascularization, and then achieving the goal of pain relief [17,40]. However, the serum levels of some chondroprotective markers showed no significant changes after treatment in this study. Many studies proved that ESWT has dose-dependent effects in different tissues. Wang et al. showed that over-dose ESWT causes deteriorating changes in osteoarthritis of knees in rats [17]. In this study, we applied a relatively lower dosage of shockwave over knees in human subjects to prevent possible complications. We believed that this dose of shockwave cannot cause a significant change of serum biologic markers.

Dr. Wang reported ESWT increased bone volume and trabecular number and decreased bone porosity significantly in osteoarthritis of the knees in rats. The BMD values also increased significantly after ESWT [23]. Osteoporosis increases the severity of cartilage damage in knee OA. In osteoporotic OA knee, the ESWT significantly improves BMD, bone-strength, subchondral-plate thickness, and bone porosity, ameliorating OA progression [41]. In the early stage OA knee, the increased bone resorption reduces the subchondral bone volume, and ESWT may halt this process, then achieving the chondroprotective effect. In our study, the bone marrow density over the tibia improved significantly after ESWT in early OA of the knees. This finding was consistent with previous reports that ESWT could slow down the progression of OA change by improving subchondral bone conditions. Compared with the HA group, the ESWT preserved the bone marrow density and even enhanced the bone marrow density over the tibial plateau. These outcomes indicated that the ESWT was better than HA for knee OA considering the subchondral bone and bone marrow density.

Limitations existed in the present study. The sample sizes are relatively smaller even though it meets the statistical requirement in power analysis. The small sample size may also explain the no difference in molecular changes in all the three groups. Besides, radiographic evaluations for osteoarthritic changes of the knees within one-year time period may not be long enough in this study protocol. There are potential biases in inter-observer evaluation. MRI is expensive, and the sensitivity is not as good as expected in the assessment for OA changes of the knee joint. Patient’s compliance in oral administration of NSAIDs required stringent study protocol. The side effects from NSAIDs are causal and the arbitrary and necessarily confirmed.

## 5. Conclusions

ESWT is an effective treatment for the symptomatic early OA knee in human subjects, and the outcomes were similar to the reported animal studies. We demonstrated comparative improvement in pain and functional scores after ESWT. The mechanical stimulation of ESWT on the subchondral bone might behave as a disease-modifying role in the progression of knee OA. In this study, ESWT posed better analgesic effect and functional score improvement in the treatment of knee OA than the HA and NSAIDs, in BMD. However, more large-scale and powerful studies are needed to prove the therapeutic effect of ESWT in patients with early knee osteoarthritis.

## Figures and Tables

**Figure 1 biomedicines-10-00202-f001:**
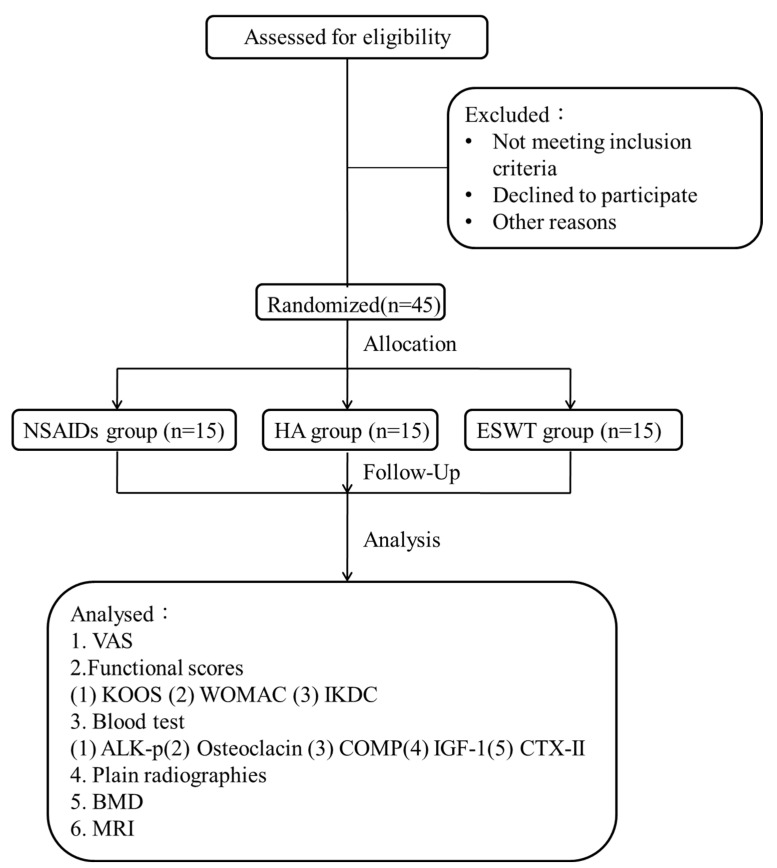
CONSORT flow diagram. NSAIDs, non-steroid anti-inflammatory drugs. HA, hyaluronic acid. ESWT, extracorporeal shockwave therapy. VAS, visual analogue scale. KOOS, knee injury and osteoarthritis outcome. WOMAC, Western Ontario and McMaster Universities Arthritis Index. IKDC, International Knee Documentation Committee. ALK-p, alkaline phosphatase. COMP, cartilage oligomeric matrix protein. IGF-1, insulin-like growth factor-1. CTX-II, C-telopeptide fragments of type II collagen. BMD, bone mineral density. MRI, magnetic resonance imaging.

**Figure 2 biomedicines-10-00202-f002:**
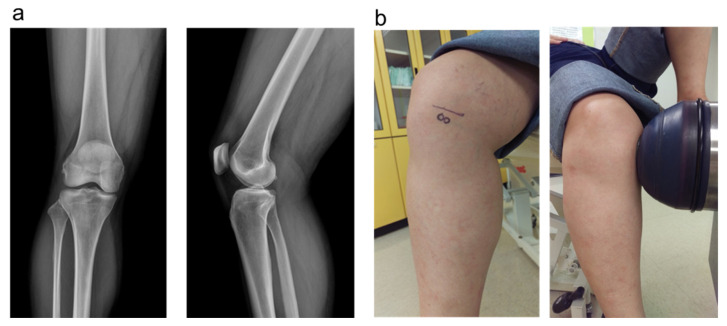
(**a**) The knee radiography showed Kellgren-Lawrence classification grade I from one of our patients. (**b**) Application of shockwave on the subchondral bone of medial tibia condyle of knee at 2.0 cm below the joint line in anteroposterior view and 2.0 cm from the medial skin edge in lateral view.

**Figure 3 biomedicines-10-00202-f003:**
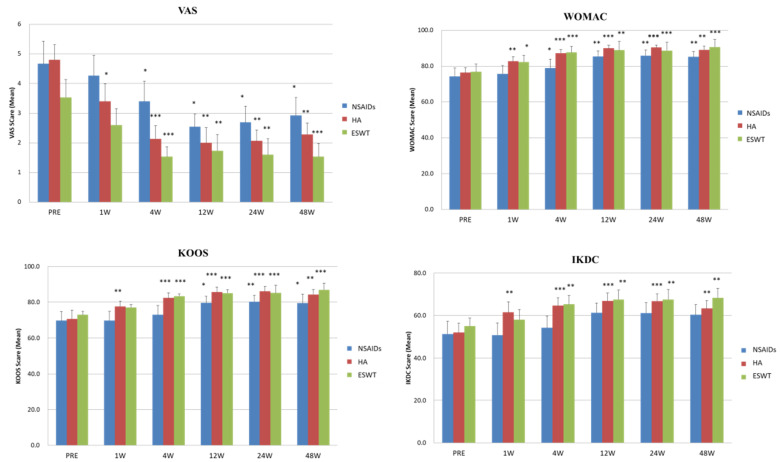
The trends of pain and functional scores improvement during our treatment course at 1, 4, 12, 24, and 48 week (W). * *p*-value < 0.05, ** *p*-value < 0.01, *** *p*-value < 0.001 as compared with the data before treatment (PRE).

**Figure 4 biomedicines-10-00202-f004:**
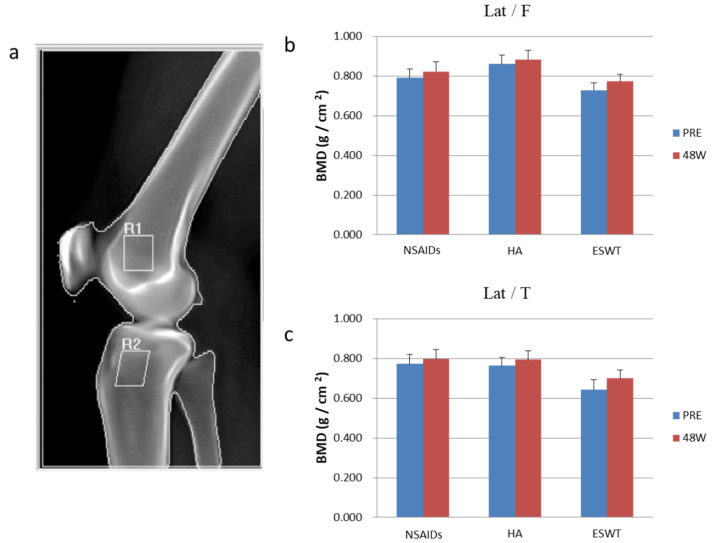
(**a**) The examined region of dual-energy X-ray absorptiometry. (**b**) The bone marrow densitometry of femur of three groups before and after treatment. (**c**) The bone marrow densitometry of tibia of three groups before and after treatment. * *p* < 0.05 as compared with data before and after treatment one-year follow-up (48 weeks). Lat/F, lateral femur. Lat/T, lateral tibia. BMD, bone mineral density. R1 and R2, region 1 and region 2.

**Table 1 biomedicines-10-00202-t001:** Demographic data of patients of three groups.

	NSAIDs	HA	ESWT	Overall
Patient numbers/Knees	15/17	15/17	15/16	45/50
Average age	49	52	54	51
(Range)	(30–63)	(33–64)	(40–60)	(30–64)
Gender (Male/Female)	1/14	2/13	1/14	4/41
Side				
Right/Left	5/8	6/7	9/5	20/20
Bilateral knees	2	2	1	5

**Table 2 biomedicines-10-00202-t002:** The results of pain scores and functional outcomes between the three groups before and after treatment.

	NSAIDs	HA	ESWT	*p*-Value *^b^	*p*-Value *^c^	*p*-Value *^d^
VAS score						
Before treatment	4.7 ± 0.8	4.8 ± 0.5	3.5 ± 0.6			
After treatment	2.9 ± 0.6	2.3 ± 0.4	1.5 ± 0.4	0.614	0.015 *	0.015 *
*p*-value *^a^	0.037 *	0.005 *	0.001 *			
KOOS score						
Before treatment	69.7 ± 5.1	70.6 ± 2.8	73.0 ± 4.1			
After treatment	79.4 ± 3.6	84.4 ± 2.2	87.0 ± 4.2	0.254	0.042 *	0.005 *
*p*-value *^a^	0.028 *	0.006 *	0.001 *			
WOMAC score						
Before treatment	74.3 ± 4.6	76.4 ± 2.7	76.9 ± 4.2			
After treatment	85.3 ± 3.0	89.2 ± 1.9	90.7 ± 4.0	0.224	0.023 *	0.006 *
*p*-value *^a^	0.005 *	0.006 *	0.001 *			
IKDC score						
Before treatment	51.3 ± 6.0	51.9 ± 4.4	55.0 ± 3.8			
After treatment	60.3 ± 4.8	63.3 ± 3.8	68.4 ± 4.4	0.697	0.325	0.24
*p*-value *^a^	0.136	0.005 *	0.005 *			

*p*-value *^a^: comparison of data before and after treatment; *p*-value *^b^: comparison of data between NSAIDs and HA; *p*-value *^c^: comparison of data between HA and ESWT; *p*-value *^d^: comparison of data between NSAIDs and ESWT. *: a *p*-value of < 0.05 was considered to be statistically significant.

**Table 3 biomedicines-10-00202-t003:** The results of blood tests between three groups before and after treatment.

	NSAIDs	HA	ESWT	*p*-Value *1	*p*-Value *2	*p*-Value *3
ALK-P						
Before treatment	63.7	67.0	65.1			
After treatment	64.6	68.4	65.3	0.367	0.298	0.362
*p*-value *	0.371	0.298	0.450			
Osteoclacin						
Before treatment	5.0	4.4	5.0			
After treatment	4.2	4.0	3.9	0.286	0.896	0.394
*p*-value *	0.221	0.397	0.001 *			
COMP						
Before treatment	165.1	185.8	173.7			
After treatment	191.1	209.8	174.2	0.028	0.107	0.357
*p*-value *	0.061	0.024 *	0.484			
IGF-1						
Before treatment	134.2	123.8	116.0			
After treatment	125.3	111.8	120.2	0.217	0.206	0.389
*p*-value *	0.122	0.077	0.212			
CTX-II						
Before treatment	0.9	1.3	1.3			
After treatment	0.8	1.1	1.2	0.058	0.76	0.102
*p*-value *	0.345	0.397	0.65			

*p*-value *: comparison of data before and after treatment; *p*-value *1: comparison of data between NSAIDs and HA; *p*-value *2: comparison of data between HA and ESWT; *p*-value *3: comparison of data between NSAIDs and ESWT. *: a *p*-value of < 0.05 was considered to be statistically significant.

## Data Availability

Not applicable.

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
