# Peer review of "Comparison of Extracorporeal Shockwave Therapy with Non-Steroid Anti-Inflammatory Drugs and Intra-Articular Hyaluronic Acid Injection for Early Osteoarthritis of the Knees"

_biomedicines, 2022, doi:10.3390/biomedicines10020202_

Round 1

Reviewer 1 Report

Dear authors.

Osteoarthrosis is one of the most common diseases of the connective tissue. Key symptoms include joint pain and stiffness.  The search for ways to minimize their negative impact is an urgent area of research. The topic touched upon in the article is relevant. The scientific content of the manuscript justifies its publication, but some additions and modifications will significantly improve the quality of the article.

Major comments:

1) L. 119-120. Does this description apply to all patients in the study or only to the last group?

2) Statistical analysis. It is known that Student's t-test is used for samples whose values are subject to a normal distribution. Has the verification of the correspondence of the samples distribution to the normal distribution been performed?

3) L. 276-278. What is the difference between experiments that can explain the pronounced significant effectiveness of exposure in your study? This information should be added.

4) L. 162-164  should be deleted.

5) Table 1. Why did women predominate in the samples?

6) Conclusions. Given the limited volume of samples, a softer statement should be made about the effectiveness of therapy.2) In the References, 31% of publications refer to 2017-2021 (the last 5 years); the remaining 69% of used sources are older than 5 years. It is recommended to increase the share of references to sources published over the last 5 years when analyzing the current state of research in the area under consideration, since this area of knowledge is rapidly developing.

Author Response

Major revision

Reviwer1

Comments and Suggestions for Authors

Dear authors.

Osteoarthrosis is one of the most common diseases of the connective tissue. Key symptoms include joint pain and stiffness.  The search for ways to minimize their negative impact is an urgent area of research. The topic touched upon in the article is relevant. The scientific content of the manuscript justifies its publication, but some additions and modifications will significantly improve the quality of the article.

Major comments:

  • 119-120. Does this description apply to all patients in the study or only to the last group?

Response: We are very grateful to the reviewer for the thoughtful comments and suggestions. The description is applied to all patients in the study and we corrected it.

  • Statistical analysis. It is known that Student's t-test is used for samples whose values are subject to a normal distribution. Has the verification of the correspondence of the samples distribution to the normal distribution been performed?

Response: Thank you so much for your comment. All values were determined whether it is subject to a normal distribution by Kolmogorov-Smirnova test. In variable factors with normal distribution, we used Student's t-test to perform statistical analysis. However, in nonparametric factors, the Wilcoxon Signed Ranks test and Mann-Whitney U test were used to perform statistical analysis.

  • 276-278. What is the difference between experiments that can explain the pronounced significant effectiveness of exposure in your study? This information should be added.

Response: Thank you so much for your comment. We described the difference between experiments as following: We applied focused ESWT in our patients and the therapeutic point was subchondral bone. The subchondral bone plays an important role in treatment of early knee OA [14]. Dr. Xu used radial ESWT and their therapeutic point was tender point of knee joints [34]. These differences might explain the different results between experiments.

4) L. 162-164  should be deleted.

Response: Thank you so much for your comment. We verified the redundant sentences.

5) Table 1. Why did women predominate in the samples?

Response: We randomized to enroll the patients and women are just predominant in the study.  

6) Conclusions.

Given the limited volume of samples, a softer statement should be made about the effectiveness of therapy.

Response: Thanks for your advice. We adjust part of our conclusion as following: The mechanical stimulation of ESWT on the subchondral bone might behave as a disease-modifying role in the progression of knee OA. In this study, in contrast to BMD, ESWT posed the same analgesic effect and functional score improvement as the HA and NSAIDs, in contrast to BMD. However, more large scale and powerful studies will need to prove the therapeutic effect of ESWT in patients with early knee osteoarthritis.

2) In the References, 31% of publications refer to 2017-2021 (the last 5 years); the remaining 69% of used sources are older than 5 years. It is recommended to increase the share of references to sources published over the last 5 years when analyzing the current state of research in the area under consideration, since this area of knowledge is rapidly developing.

Response: Thank you very much. We verified the references and added more references in last 5 years.

Reviewer 2 Report

 The present study compared the efficacy of oral 22 NSAIDs, HA injection and noninvasive ESWT for early OA of the knee. Forty-five patients with 23 early knee OA were randomized into three groups. The results demonstrated that ESWT was an effective and alternative therapy than HA and NSAIDs for early osteoarthritis of the knees.

Methodology

  • The study is interesting but the main problem is the sample size, which, in the limitations of the study, the authors indicate that it is small. It would be interesting to indicate in the statistical methodology the sample size necessary for the study to have sufficient statistical power.
  •       Analytical determinations provide little information to the study and are probably not necessary
  • Discussion      
  • The discussion is excessively long. The authors should further discuss the cellular mechanisms that explain the benefits of shock waves and focus on the differences between their study and those previously published. 

Author Response

Reviewer2

Comments and Suggestions for Authors

 The present study compared the efficacy of oral 22 NSAIDs, HA injection and noninvasive ESWT for early OA of the knee. Forty-five patients with 23 early knees OA were randomized into three groups. The results demonstrated that ESWT was an effective and alternative therapy than HA and NSAIDs for early osteoarthritis of the knees.

Methodology

The study is interesting but the main problem is the sample size, which, in the limitations of the study, the authors indicate that it is small. It would be interesting to indicate in the statistical methodology the sample size necessary for the study to have sufficient statistical power.

      Analytical determinations provide little information to the study and are probably not necessary

Response: Thank you so much for your comment. The sample size of n = 45 was estimated by Power analysis software (G*Power, version 3.1.9.2) with the Post hoc test (two-tailed test [α, 0.05; β, 0.467]) and estimated a sample size of 45 participants (15 participants per group). The output data is including noncentrality parameter (5.866), critical t (2.022), Df (39.107) and Power (1- β err prob) (> 0.9).

Discussion     

The discussion is excessively long. The authors should further discuss the cellular mechanisms that explain the benefits of shock waves and focus on the differences between their study and those previously published.

Response: We appreciate reviewer’s comment. We deleted part of section to simplify our discussion. We compared the results of ESWT on early knee osteoarthritis in our study with previous other studies in section 2. We further discussed the therapeutic effect of ESWT on OA knees with NSAIDs and HA in section 3 and 4. The possible cellular mechanisms of ESWT on OA knees were added in introduction and section 5 of the discussion. However, we applied relative low dosage of ESWT on human bodies in this study, so the serum biologic markers had no significant change. We still discuss some possible mechanisms previous studies reported. 

Reviewer 3 Report

Dear Authors,

This is a very interesting original article. It was a prospective, single-center study. The study is well designed according to STROBE guidelines, the methods and results are well presented. The discussion is interesting and well written.

Nevertheless, I have some suggestions to improve the paper:

  1. I suggest to consider add some information about genetics in the pathogenesis and biomarkers of OA in the introduction section, especially that it was one of the end-points. 
  2. Some information about new intra-articular methods e.g. PRP effect on the joint (e.g. doi: 10.3390/ijms22115492.) can be mentioned in the introduction or discussion section.
  3. Authors could add the information about new procedures treating the bone marrow edema associated with OA (e.g. doi: 10.1016/j.eats.2020.07.023) in the introduction section or discussion section.
  4. The sample size calculation should be added in the statistical analysis section.
  5. The CONSORT flow chart should be added in the methods section.

Author Response

Reviewer3

Comments and Suggestions for Authors

Dear Authors,

This is a very interesting original article. It was a prospective, single-center study. The study is well designed according to STROBE guidelines, the methods and results are well presented. The discussion is interesting and well written.

Nevertheless, I have some suggestions to improve the paper:

I suggest to consider add some information about genetics in the pathogenesis and biomarkers of OA in the introduction section, especially that it was one of the end-points.

Response: We described the information about pathogenesis and biomarkers of OA in the section of introduction as follows: Many inflammatory cytokines (interleukin-1β, IL1-β; interleukin-6, IL-6; and tumor necrosis factor-α, TNF-α) and pathological factors (matrix metalloproteinase-13, MMP-13; vascular endothelial growth factor, VEGF) were released to synovial fluid of knee OA[2]. In addition, inflammation–induced Transforming growth factor beta (TGFβ), connective tissue growth factor (CTGF), Bone Morphogenetic Protein-2 (BMP-2), BMP-5 and BMP-6 were caused the synovial and articular cartilage fibrosis and death in severe knee OA[3,4].

Some information about new intra-articular methods e.g. PRP effect on the joint (e.g. doi: 10.3390/ijms22115492.) can be mentioned in the introduction or discussion section.

Response: We appreciate reviewer’s comment. We described the effect of intra-articular PRP and cited the reference in the section of introduction as follows: For example, platelet-rich plasma (PRP) is an autologous concentration of a patient's own platelets, and can be fast preparation to inject into knee OA of patient by intra-articular injection. The mechanism of PRP is clear understanding to anti-inflammation, stimulated chondrocyte proliferation, and tissue regeneration for OA in clinical trial and animal studies[7].

  1. Szwedowski, D.; Szczepanek, J.; Paczesny, Ł.; Zabrzyński, J.; Gagat, M.; Mobasheri, A.; Jeka, S. The Effect of Platelet-Rich Plasma on the Intra-Articular Microenvironment in Knee Osteoarthritis. International Journal of Molecular Sciences 2021, 22, 5492, doi:10.3390/ijms22115492.

 Authors could add the information about new procedures treating the bone marrow edema associated with OA (e.g. doi: 10.1016/j.eats.2020.07.023) in the introduction section or discussion section.

Response: We appreciate reviewer’s comment. We described a new method for bone marrow edema in OA as follows: Surgical treatments include arthroscopic debridement and release, osteotomy and/or joint replacement. In addition, a new method which is named osteo-core plasty is a minimally invasive procedure to use for subchondral bone marrow lesion of knee OA[8].

  1. Szwedowski, D.; Dallo, I.; Irlandini, E.; Gobbi, A. Osteo-core Plasty: A Minimally Invasive Approach for Subchondral Bone Marrow Lesions of the Knee. Arthroscopy Techniques 2020, 9, e1773-e1777, doi:10.1016/j.eats.2020.07.023.

The sample size calculation should be added in the statistical analysis section.

Response: Thank you so much for your comment. The sample size calculation was added in the section of “2.2. The patients” of Material and Methods as follows: The sample size of n = 45 was estimated by Power analysis software (G*Power, version 3.1.9.2) with the Post hoc test (two-tailed test [α, 0.05; β, 0.467]) and estimated a sample size of 45 participants (15 participants per group). The output data is including noncentrality parameter (5.866), critical t (2.022), Df (39.107) and Power (1- β err prob) (> 0.9).

The CONSORT flow chart should be added in the methods section.

Response: Thank you so much for your comment. The CONSORT flow chart was shown in the figure 1.

Reviewer 4 Report

I read with interest the article about the comparison of extracorporeal shockwave therapy with NSAIDs and intra-articular HA injection for early OA of the Knees. Due to the prevalence of OA in society and the limited opportunities for effective treatment, every research on this subject is valuable for clinical practice. In the text presented for review, there are minor spelling mistakes, eg in the abbreviations of NSAIDs, etc. Unfortunately, the authors included a very small group of patients in the study, additionally, the observation of the effects of the study procedures was very short. In fact, the authors indicated significant improvements in pain and function, the results of ELISA analysis of the serum levels of ALK-P, osteoclacin, COMP, IGF-1 and CTX-II showed no significant differences before and after treatment in the groups studied . There were also no differences of MRI parameters before and after treatment, however the authors reported that the BMD significantly increased in proximal tibia only after ESWT treatment. It should be noted that the authors did not report BMD values ​​before and after treatment - the only data are in Fig 3, where it appears that the BMD in the ESWT group is lower at baseline than in the other groups. Unfortunately, no statistical comparisons are given for the baseline BMD, therefore the significance cannot be assessed. I believe that significant clinical improvement in the early stage of OA can be achieved with a variety of methods, which does not always mean that disease progression is objectively inhibited. A small group of patients and a short observation period certainly cannot provide objective evidence of high efficacy (apart from the described analgesic effect) of ESWT in the long term. I consider the study too preliminary to publish in a journal of as high a quality as Biomedicines.

Author Response

Reviewer4

Comments and Suggestions for Authors

I read with interest the article about the comparison of extracorporeal shockwave therapy with NSAIDs and intra-articular HA injection for early OA of the Knees. Due to the prevalence of OA in society and the limited opportunities for effective treatment, every research on this subject is valuable for clinical practice. In the text presented for review, there are minor spelling mistakes, eg in the abbreviations of NSAIDs, etc. Unfortunately, the authors included a very small group of patients in the study, additionally, the observation of the effects of the study procedures was very short. In fact, the authors indicated significant improvements in pain and function, the results of ELISA analysis of the serum levels of ALK-P, osteoclacin, COMP, IGF-1 and CTX-II showed no significant differences before and after treatment in the groups studied. There were also no differences of MRI parameters before and after treatment, however the authors reported that the BMD significantly increased in proximal tibia only after ESWT treatment. It should be noted that the authors did not report BMD values ​​before and after treatment - the only data are in Fig 3, where it appears that the BMD in the ESWT group is lower at baseline than in the other groups. Unfortunately, no statistical comparisons are given for the baseline BMD, therefore the significance cannot be assessed. I believe that significant clinical improvement in the early stage of OA can be achieved with a variety of methods, which does not always mean that disease progression is objectively inhibited. A small group of patients and a short observation period certainly cannot provide objective evidence of high efficacy (apart from the described analgesic effect) of ESWT in the long term. I consider the study too preliminary to publish in a journal of as high a quality as Biomedicines.

Response: Thank you so much for your comment. The baseline of BMD was calculation in the three groups by Oneway method as well as showed no difference and was confidence.

The sample size calculation was added in the section of “2.2. The patients” of Material and Methods as follows: The sample size of n = 45 was estimated by Power analysis software (G*Power, version 3.1.9.2) with the Post hoc test (two-tailed test [α, 0.05; β, 0.467]) and estimated a sample size of 45 participants (15 participants per group). The output data is including noncentrality parameter (5.866), critical t (2.022), Df (39.107) and Power (1- β err prob) (> 0.9).

Round 2

Reviewer 1 Report

Dear Authors

My comments are taken into account.

Reviewer 2 Report

The answers of the authors are correct

Reviewer 4 Report

The study may be published in present form.